# Farming Practices, Biosecurity Gaps, and Genetic Insights into African Swine Fever Virus in the Iringa and Ruvuma Regions of Tanzania

**DOI:** 10.3390/ani15071007

**Published:** 2025-03-31

**Authors:** Agathe Auer, Anderson Samwel Yohana, Tirumala B. K. Settypalli, Raphael Sallu, Jelly Chang’a, Stella Bitanyi, Stella Gaichugi Kiambi, Irene K. Meki, William G. Dundon, Artem Metlin, Andriy Rozstalnyy, Geofrey Hallan Mbata, James Anset Okachu, Henry Magwisha, Sauda Ally Hamis, Jeremia Theodos Choga, Stela Lucas Chalo, Joshua Kimutai, Gerald Misinzo, Solomon Wilson Nong’ona, Joseph Edmund Lyimo, Charles E. Lamien

**Affiliations:** 1Animal Production and Health Laboratory (APHL) at the Joint FAO/IAEA Centre of Nuclear Techniques in Food and Agriculture, Department of Nuclear Sciences and Applications, International Atomic Energy Agency (IAEA), Wagramer Strasse 5, P.O. Box 100, A-1400 Vienna, Austria; 2Food and Agriculture Organization of the United Nations (FAO), 00153 Rome, Italy; 3Tanzania Veterinary Laboratory Agency Iringa (TVLA), Iringa P.O. Box 290, Tanzania; 4Food and Agriculture Organization of the United Nations (FAO), Emergency Centre for Transboundary Animal Diseases, Dar Es Salaam 14111, Tanzania; 5Zonal Veterinary Investigation Centre (ZVC), Southern Highland Zone, Iringa P.O. Box 290, Tanzania; 6Food and Agriculture Organization of the United Nations (FAO), Emergency Centre for Transboundary Animal Diseases, Nairobi 00100, Kenya; 7Department of Microbiology, Parasitology and Biotechnology, College of Veterinary Medicine and Biomedical Sciences, Sokoine University of Agriculture, P.O. Box 3019, Morogoro 67152, Tanzania

**Keywords:** ASFV, biosecurity, molecular characterization, Tanzania, pig farming

## Abstract

African Swine Fever Virus (ASFV) poses a severe threat to pig farming globally. This study investigates farming practices and ASFV genotype II distribution in the Iringa and Ruvuma Regions of Tanzania, focusing on biosecurity gaps and molecular characterization. The study attempted to establish the ASFV infection status and the standards of biosecurity status in a select group of holdings. Therefore, a total of 205 clinical specimens were collected from 120 farm holdings, resulting in the confirmation of 21 ASFV cases from 14 holdings, and the detection of porcine circovirus-2 (PCV-2) co-infection in some cases. These findings emphasize the need for targeted biosecurity interventions and surveillance to mitigate ASF outbreaks in endemic regions. Phylogenetic analysis showed genetic uniformity among isolates, underlining the persistent dominance and stability of genotype II in the region.

## 1. Introduction to ASF and Current Disease Situation in Tanzania

African Swine Fever (ASF) remains one of the most significant challenges to global pig farming, with genotype II identified as a highly virulent strain, causing case fatality rates exceeding 90% and compounded by the lack of an effective vaccine [1,2,3,4]. ASF virus (ASFV) is a large DNA arbovirus and the sole member of the Asfarviridae family, genus Asfivirus [5,6]. There are currently 24 recognized genotypes of ASFV based on analysis of the C-terminal of the p72 gene [7,8]. In Tanzania, ASF has had a particularly devastating impact on the Southern Highlands, a region that contributes nearly half (47%) of the national domestic pig population. The Southern Highlands, encompassing Iringa, Mbeya, Songwe, Njombe, and Ruvuma regions, are home to an estimated 1.5 million pigs. In Tanzania, the estimated pig population is 3.2 million and represents 4.5% of the national quadruped meat-producing livestock population (Table 1). With an average of six pigs per holding, this results in approximately 250,000 pig holdings across the Southern Highlands [9].

ASF was first described in Tanzania in 1914 and officially reported to the WOAH in 1962. Historically, outbreaks were sporadic, often separated by long intervals. Major epidemics occurred in 1987 and 1988, impacting the Southern and Northern highlands, particularly Mbeya, Arusha, and Kilimanjaro [10,11,12]. However, since 2000, ASF outbreaks have become more frequent. A major turning point was the 2010 epidemic, which originated in Kyela, Mbeya, and spread to other regions, including Morogoro and Dar-es-Salaam (Figure 1). This outbreak, traced to Karonga District in Malawi, was caused by ASFV genotype II, marking its first detection in Tanzania [13]. Meanwhile, the concurrent epidemic in the Northern Highlands was attributed to genotype X [14]. Tanzanian ASF outbreaks have been linked to multiple ASFV genotypes, including II, IX, X, XV, and XVI [15,16]. Before 2010, genotypes X, XV, and XVI were the primary ones circulating in the country. Between 2015 and 2017, genotypes II and, to a lesser extent, IX and X were responsible for most ASF outbreaks in Tanzania [14,17,18].

Since June 2023, the Tanzania Veterinary Laboratory Agency (TVLA) in Iringa has been actively engaged in field epidemiological studies to detect ASFV using qPCR and better understand the real-world biosecurity practices implemented on small farm holdings. As part of this effort, a short questionnaire was introduced to capture insights into on-farm biosecurity measures, enabling an assessment of the practices employed in managing ASF outbreaks. These activities were supported by a participatory training course conducted by the FAO, focusing on ASF detection, diagnosis, and biosecurity protocols during sampling. With additional backing from the FAO and the Joint FAO/IAEA Centre, TVLA Iringa has enhanced its diagnostic capacity to address the escalating ASF cases in the Southern Highlands. This study investigates the circulating ASFV genotypes in the Iringa and Ruvuma regions of Tanzania while looking into biosecurity gaps and management practices that contribute to virus transmission in small farm holdings. Moreover, it assesses the potential for co-infections with other pathogens —such as porcine circovirus-2 (PCV-2), porcine circovirus-3 (PCV-3), porcine parvovirus 1 (PPV1), pseudorabies virus (PrV), *Erysipelothrix rhusiopathiae*, and *Salmonella* spp.—which could complicate ASF management and control efforts.

## 2. Materials and Methods

### 2.1. Sampling Strategy

Between July 2023 and June 2024, ASF sampling was conducted in Tanzania’s Southern Highlands, focusing on the regions of Iringa, Mbeya, and Ruvuma. These areas were selected due to their high domestic pig population density and history of ASF outbreaks. Following training provided by the FAO, TVLA Iringa was equipped and trained in field sampling and the detection of ASF. Equipped with this capacity, TVLA teams carried out sampling operations in the field. Farm holdings were selected based on farmers’ willingness to participate, as some declined access due to personal cultural or religious beliefs or concerns. Sampling mainly targeted farm holdings where farmers requested assistance for ASF detection. A total of 120 farms were visited and 205 biological samples (whole blood) were collected from domestic pigs exhibiting signs of illness, such as lethargy or other clinical signs. On average, two animals were sampled per holding. Each sample was assigned a unique identifier, ranging sequentially from Tan_ASF2024_01 to Tan_ASF2024_205.

### 2.2. Data Collection and Biosecurity Assessment

Data collection was conducted via questionnaires (Appendix A). The questionnaires were administered after voluntary written consent of the farmers. The questionnaire was designed to gather detailed information on several key variables: farm holding location, domestic pig demographics, types of samples collected, and GPS data for spatial analysis. It also assessed disease signs, progression, and farming systems (e.g., feeding practices, housing practices). Questions on biosecurity measures and additional observations on signs at the time of sample collection and co-infections were included to provide deeper insights into the health status of the pigs and the effectiveness of implemented biosecurity protocols if any.

### 2.3. Laboratory Analysis

Samples were analyzed using DNA Blood and Tissue Kit (QIAGEN, Hilden, Germany) and qPCR to confirm ASF virus presence, and any remaining samples (DNA) were securely stored at −20 °C for future analysis. ASFV was detected using King et al.’s (2003) [19] protocol and validated master mixes [20]. Auer et al. (2022) [20] analyzed master mixes by assessing sensitivity, specificity, and reproducibility using ASFV-positive and ASFV-negative DNA samples, comparing performance based on Cq values, limit of detection, and concordance with reference methods.

### 2.4. Detection of Co-Infections and Molecular Characterization

#### 2.4.1. Detection of ASFV and Other Pathogens

ASFV-positive and suspect DNA samples, initially tested as described above at the TVLA, were sent to the Animal Production and Health Laboratory (APHL), at the Joint FAO/IAEA Centre, in Seibersdorf, Austria, for further analysis (Appendix A). ASFV was detected using the method described by King et al. (2003) [19]. The samples were then screened for other pathogens using the following two protocols:(1)An in-house multiplex assay was employed to screen for *Erysipelothrix rhusiopathiae* and *Salmonella* spp., in addition to ASFV, as per protocols developed by the APHL.(2)A second one for PCV-2, PCV-3, PPV1, and PrV screening using the methodology described by Chen et al. (2023) [21].

Co-infection testing was conducted using the available DNA, with a focus on differential diagnosis to distinguish other pathogens from ASFV.

#### 2.4.2. Molecular Characterization of ASFV Genes

Molecular characterization of ASFV-positive samples was conducted through the amplification and sequencing of five genomic regions:B646L Gene (p72 Protein): the C-terminus of the B646L gene was amplified and sequenced to determine ASFV genotype following the method outlined by Bastos et al. (2003) [22].E183L Gene (p54 Protein): the complete E183L gene was amplified according to the protocol established by Gallardo et al. (2009) [23].CD2v Protein Sequences: the CD2v fragment was amplified using the methodology of Malogolovkin et al. (2015) [24].Central Hypervariable Region of B602L Gene (CVR): the CVR of the B602L gene was analyzed following the protocol by Nix et al. (2006) [25].Intergenic Region (ECOA): the intergenic region between the I73R and I329L genes was amplified to evaluate genetic variation [26].

All PCR products were purified and sequenced using the Sanger method. The obtained sequences were assembled using Vector NTI software v11.5 (Invitrogen), then subjected to multiple sequence alignments and phylogenetic analyses on MEGA11 with ASFV reference sequences retrieved from the GenBank, to assess the genetic relationships and determine serogroup classifications.

### 2.5. Data Analysis

Neighbor-joining, Minimum Evolution and maximum likelihood trees were constructed based on p72, p54 (nucleotide), and CD2v (amino acid) sequences, respectively, using appropriate evolutionary models. Bootstrap values greater than 70% were reported to indicate support for the branches. Multiple sequence alignments and analysis of the ECOA intergenic region were conducted using BioEdit v7.2.5 to evaluate similarities and identify specific deletions or mutations relevant to the study. The CVR amino acid tetramers and profiles were deduced by translating the nucleotide sequences to protein sequences using Python code (version 3.7) on the Spyder IDE.

## 3. Results

### 3.1. PCR Results

A total of 205 samples were collected for ASFV screening from 120 farm holdings across the Southern Highland Zone. Among these, 21 samples tested positive within the Ruvuma and Iringa regions. The geographic distribution of the ASF sample collection sites is illustrated in Figure 2. Sampled districts are highlighted in green, while wards with confirmed ASF-positive samples are marked in red, indicating specific areas where the virus was detected. To provide geographical context and support spatial analysis of ASF spread, water bodies are depicted in blue. The Cq values for ASFV-positive samples ranged from 19.23 to 29.2. Notably, six samples were found to be co-infected with PCV-2. Other swine pathogens including PCV-3, PPV1, PrV, *E. rhusiopathiae*, and *Salmonella* spp. were not detected.

### 3.2. Molecular Characterization of Tanzanian ASFV Isolates

Amplification and sequencing of the targeted ASFV genes, the p72, p54, CD2v, CVR, and ECOA fragments, were successful in 17, 11, 17, 14, and 13 samples, respectively. Firstly, the multiple sequence alignments of each fragment showed 100% nucleotide similarity among all sequenced ASFV samples from Tanzania. For comparative analysis, only six representative samples sequenced for all five genes were further analyzed and submitted to GenBank under accession numbers PQ800626–PQ800655. Phylogenetic analysis based on both the p72 and p54 nucleotide sequences clustered the Tanzanian 2024 ASFV isolates within genotype II (Figure 3A,B).

In addition, phylogenetic analysis based on the CD2v amino acid sequences clustered all Tanzanian ASFV isolates in Serogroup 8 (Figure 4).

Following the translation of the CVR sequences into amino acids and coding to the corresponding signature, the Tanzania ASFV samples revealed 10 tetrameric repeats (BNDBNDBNAA). Finally, the analysis of the ECOA region showed a deletion of a nucleotide repeat sequence (GAATATATAG) and A/G substitution compared to other ASFV genotype II isolates (Figure 5). The nucleotide substitution/deletion in the ECOA sequences in the Tanzanian samples, shown in Figure 5, are located in the I73R–I329L intergenic spacer and not the coding regions; therefore, there was no need to analyze the amino acid sequence. The analysis of the Tanzanian ASFV 2024 isolates based on the five targeted genes clearly indicated that they are genotype II, and none of the genes classified the isolates in any other ASFV genotype, which would imply recombination.

### 3.3. Questionnaire

All 120 holdings visited had herds of 4–35 domestic pigs housed in confinement. None of the animals were raised free-range, as all owners protected their animals from theft by keeping them confined. Nearly 90% of farmers did not restrict visitor access, and 95% failed to disinfect equipment or practice basic hygiene protocols such as footbaths or quarantine measures for new domestic pigs. More than two-thirds of farmers (80%) relied on untreated swill from households and restaurants—a practice particularly prevalent in Iringa and Ruvuma. Knowledge gaps among farmers compounded these risks. While 65% could identify ASF, risky practices were common, with 50% of farmers selling or slaughtering sick domestic pigs and 10% disposing of carcasses in open fields during outbreaks. Veterinary services were found to be insufficient, with 70% of respondents citing delays or unavailability. Only 5% of farmers had received formal training in biosecurity and ASF prevention (Figure 6).

## 4. Discussion

This study provides insights into ASFV in the Iringa and Ruvuma regions of Tanzania, with findings based on convenience sampling conducted in response to farmer requests for assistance. A limitation of this study is that the findings underscore ongoing ASFV circulation but do not reflect the overall prevalence or incidence of ASFV in the Southern Highlands, as the sampling was not designed to be representative of the broader pig population. With approximately 250,000 pig holdings in the Southern Highlands, only 0.048% (120 farm holdings) were sampled over the course of a year. Instead, the results illustrate patterns associated with reported clinical cases and biosecurity gaps in the sampled farm holdings. The detection of ASFV in multiple farm holdings in Ruvuma and Iringa reflects the high-risk farming practices observed in these regions, such as reliance on untreated swill feeding (80% of holdings visited) and lack of visitor restrictions (90%). These findings emphasize the need for targeted biosecurity interventions. Recent research indicates a strong correlation between higher biosecurity scores, particularly internal measures, and improved health and production metrics, such as average daily weight gain [27]. For ASFV-endemic areas like Tanzania, measures such as proper cleaning and disinfection are critical to mitigating pathogen spread and improving outcomes.

The low rate of formal biosecurity training among farmers (5%) and risky behaviors, such as selling sick pigs (50%), further contribute to infection cycles, perpetuating the disease and highlighting the importance of farmer education and community-based interventions [28,29]. Biosecurity measures were poor in all of the farm holdings that were visited and sampled. Additionally, as holdings were only sampled on a single occasion and there was no follow-up testing (e.g., weekly testing and serological analysis using ELISA), those farm holdings from which samples tested qPCR-negative cannot be considered to be free of infection. These findings confirm that ASFV remains endemic in the region.

To better understand the relationship between biosecurity measures and ASFV prevalence in Tanzania’s Southern Highlands, future studies should incorporate quantitative biosecurity scoring systems, such as Biocheck.UGent™, to standardize and evaluate external and internal biosecurity measures across farms. Advanced detection methods, like nanopore metagenomic sequencing, could enhance pathogen surveillance by quantifying viral loads and identifying co-circulating pathogens more comprehensively than qPCR alone [30].

This study demonstrates that all sequenced ASFV isolates from this study are identical across all targeted genes and fragments, clustering in ASFV genotype II and Serogroup 8, with consistent CVR and ECOA profiles. These findings align with earlier reports of genotype II’s introduction to Tanzania in 2010 and its subsequent dominance in outbreaks [13,15,16,17,31,32] Yona et al., (2017). No evidence of other genotypes, such as IX, X, XV, or XVI, which were historically linked to earlier outbreaks, was found in this study [16]. This further supports the conclusion that genotype II has become the predominant strain in the Southern Highlands and is consistent with the broader epidemiological understanding of its dominance across Eastern and Southern Africa. The correlation between genotype II and Serogroup 8 has been observed globally, linking these strains to highly virulent outbreaks with more than 90% lethality [32,33]. Given the association of genotype II with high virulence, immediate steps should include targeted biosecurity training, especially in regions like Ruvuma and Iringa where positive cases were concentrated. Rapid diagnostic tools should also be deployed to high-risk areas to enable early detection and containment [34,35].

Co-infections with PCV-2 were identified, potentially compromising host immunity and exacerbating clinical signs [36,37,38]. No other co-infections were detected. The presence of PCV-2 highlights the potential role of opportunistic pathogens in exacerbating the impact of ASFV; however, its role remains secondary. The absence of other pathogens, including *E. rhusiopathiae*, *Salmonella* spp., PCV-3, PPV1, and PrV, suggests limited involvement of these agents in the sampled population. This could be attributed to effective regional control measures or the dominance of ASFV, which may suppress the detection or expression of other infections. PCV-2 has previously been identified in Tanzania [39].

## 5. Conclusions

Tailored control strategies must integrate genotype II’s epidemiological dominance, enhanced diagnostic tools, and biosecurity training to mitigate risks [29,39,40,41]. Strengthened collaboration among veterinarians, local authorities, and farmers is vital to improving emergency responses, building trust in veterinary services, and ensuring compliance with disease control measures [42,43,44,45,46]. This study highlights the practical challenges and provides targeted recommendations to strengthen disease control and biosecurity practices in ASFV-endemic regions like Ruvuma and Iringa in Tanzania. Further research should explore co-infections’ broader impacts on lethality and altered disease dynamics to deepen the understanding of ASFV pathogenesis under field conditions. 

## Figures and Tables

**Figure 1 animals-15-01007-f001:**
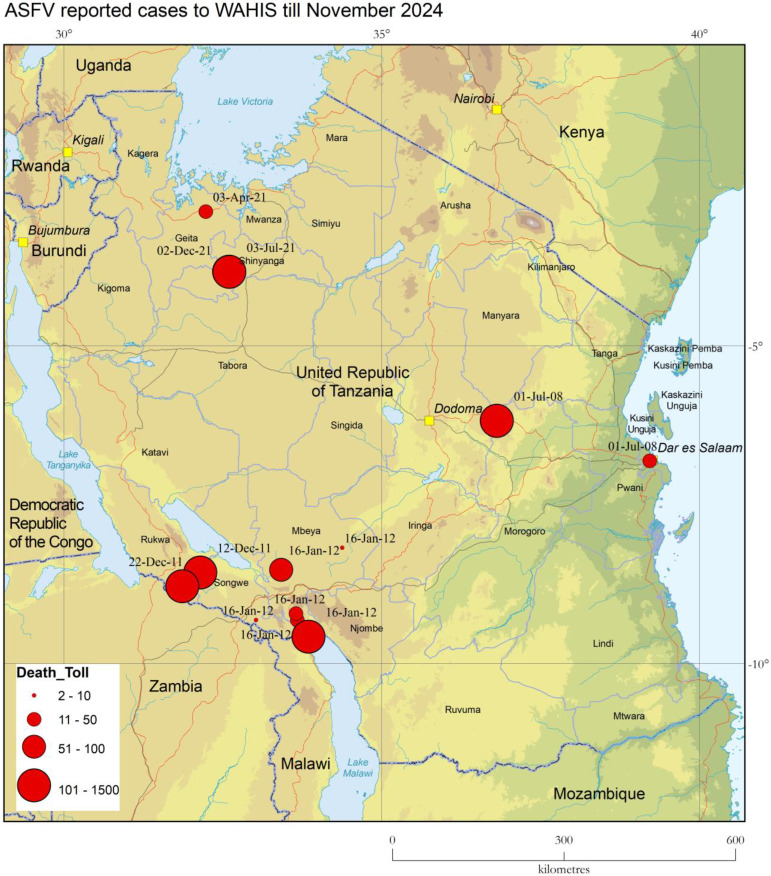
Distribution of ASF confirmed cases in Tanzania from 2008 to 2024 as reported by WAHIS, indicating the number of succumbed animals and end date of the outbreak. The yellow square indicates the capitals on this map.

**Figure 2 animals-15-01007-f002:**
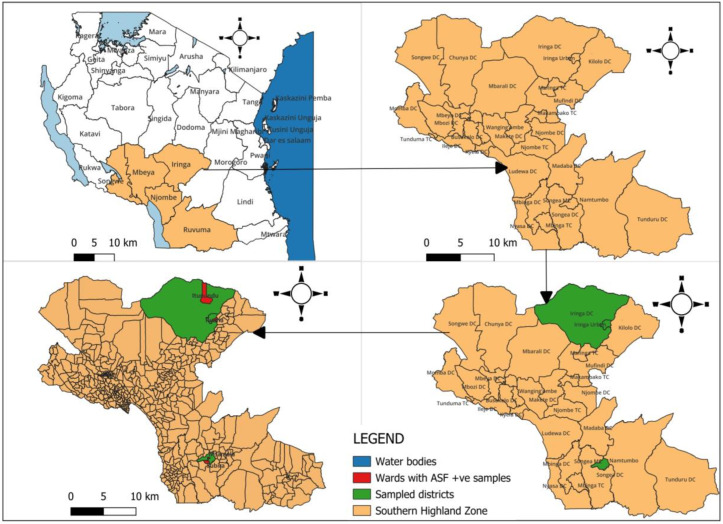
Geographical distribution of African Swine Fever (ASF) cases in Tanzania’s Southern Highland Zone (orange). The map highlights sampled districts (green) and wards with ASF-positive samples (red) within this livestock-intensive region.

**Figure 3 animals-15-01007-f003:**
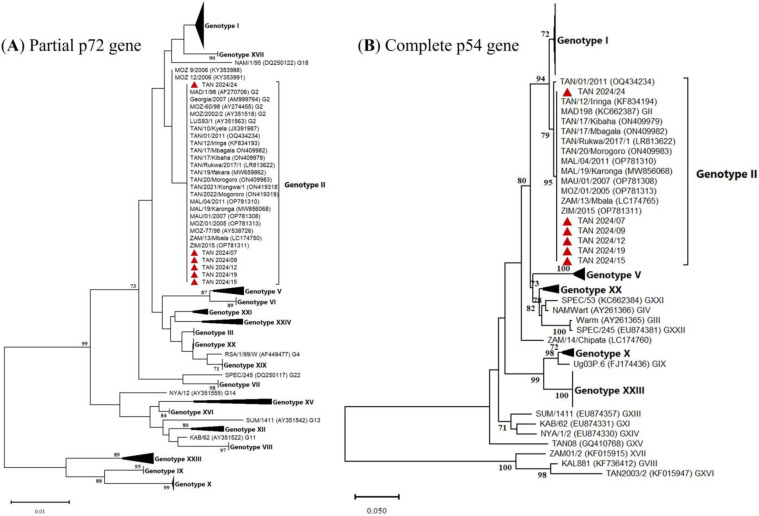
Phylogenetic trees based on (**A**) partial p72 gene nucleotide sequences, inferred using the neighbor-joining method with evolutionary distances computed using the maximum composite likelihood method and (**B**) the complete p54 gene nucleotide sequences based on Minimum Evolution method and evolutionary distances computed using the Kimura two-parameter method. The 2024 ASFV isolates from Tanzania are highlighted in red triangles. Only bootstrap values greater than 70% are shown.

**Figure 4 animals-15-01007-f004:**
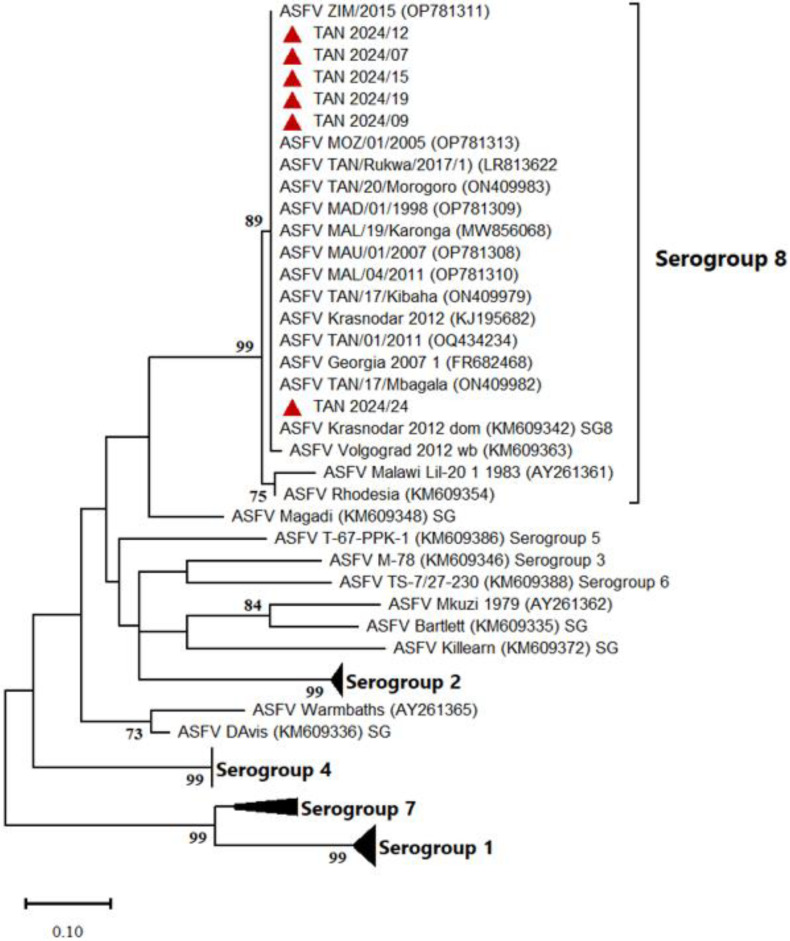
Maximum likelihood tree based on the partial amino acid sequence of the CD2v protein, showing the relationship between the 2024 ASFV isolates from Tanzania (in red triangles) and representatives of known ASFV serogroups as well as ASFVs clustering outside the eight established serogroups. The General Reversible Chloroplast model Gamma distribution was used. Only bootstrap values greater than 70% are shown.

**Figure 5 animals-15-01007-f005:**
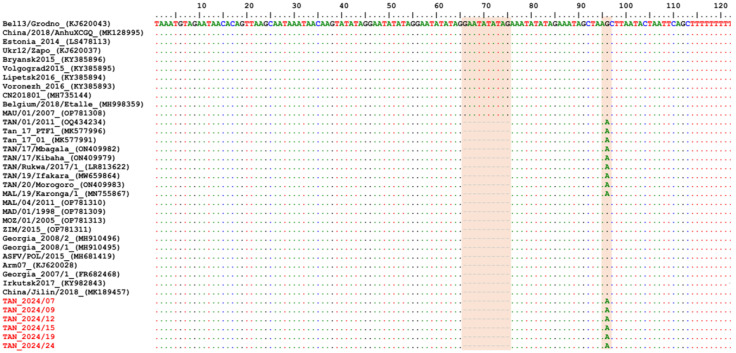
Multiple sequence alignments of the ECOA intergenic region between I73R and I329L genes of the Tanzanian ASFV GII samples (in red) in comparison to representatives of known genotype II isolates. The highlighted regions show a deletion of a nucleotide internal repeat sequence (GAATATATAG) and a G/A Single-Nucleotide Polymorphism. The dots indicate the identical nucleotides in the alignment. In the figure, the nucleotides are color-coded as follows: A is shown in green, G is shown in black, T is shown in red, and C is shown in blue.

**Figure 6 animals-15-01007-f006:**
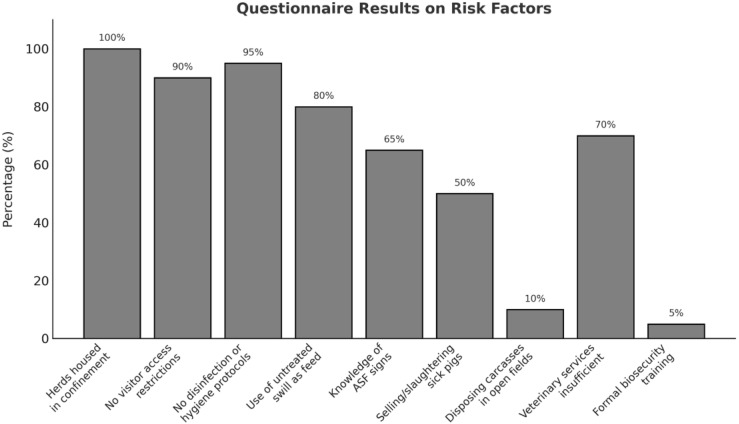
Assessment of risk factors on domestic pig holdings based on questionnaire results.

**Table 1 animals-15-01007-t001:** Key data on domestic pig farming and ASF in the Southern Highlands, Tanzania.

Category	Details
**Estimated Domestic Pig Population**	1.5 million (of 3.2 million nationally)
**ASF Outbreak Years**	2010, 2011, 2014, 2015, 2016–2020, 2022
**Pig Population Density**	Mbeya: 7.2 pigs/km^2^, Ruvuma: 6 pigs/km^2^
**Main ASF Virus Genotype**	Genotype II

## Data Availability

The data presented in this study are available upon request from the corresponding author. Field sampling and laboratory analyses were conducted in collaboration with the Tanzania Veterinary Laboratory Agency (TVLA), the Food and Agriculture Organization of the United Nations (FAO), and the Animal Production and Health Laboratory (APHL) at the Joint FAO/IAEA Centre of Nuclear Techniques in Food and Agriculture, Department of Nuclear Sciences and Applications, International Atomic Energy Agency (IAEA). Data sharing must comply with institutional policies.

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
