# Peer review of "Farming Practices, Biosecurity Gaps, and Genetic Insights into African Swine Fever Virus in the Iringa and Ruvuma Regions of Tanzania"

_animals, 2025, doi:10.3390/ani15071007_

Round 1

Reviewer 1 Report

Comments and Suggestions for Authors

The entire text contains limited information and can only serve as a informative supplement. Thus, it should be suggested that some info must added before publication below.

(1)If the genotype of PCV2 is suggested to analyze, the results will be better for readers.

(2)Sampling area compensation fairness refers to two small regions in this country that are far apart. Overall evaluation, from the sampling area perspective, it cannot represent the level of the entire country or region.

(3)PRRSV in the samples should be tested for this study?

(4)Blood collection is prone to contamination, why not collect samples such as oral or anal samples and expand the sampling area.

Author Response

thank you for your valuable feedback, which has helped us improve the clarity and rigor of the study. We hope we have fully addressed your comments (1-4).

(1)If the genotype of PCV2 is suggested to analyze, the results will be better for readers.

The focus of this study was ASFV, and other pathogens were tested primarily to assess co-infections rather than for detailed molecular characterization. While PCV2 genotyping could provide additional insights, it was beyond the scope of this study.

(2)Sampling area compensation fairness refers to two small regions in this country that are far apart. Overall evaluation, from the sampling area perspective, it cannot represent the level of the entire country or region.

The reviewer is correct that samples were collected only from Ruvuma and Iringa, which do not fully represent the epidemiological situation of the entire country. This limitation was acknowledged and thus changed in the manuscript, specifically in the title, simple summary (line 23), and discussion (lines 248–256), where we stated that the sampled regions do not necessarily reflect the national or regional situation.

(3)PRRSV in the samples should be tested for this study?

 We were unable to check for the presence of PRRSV since we were provided with DNA samples only.

(4)Blood collection is prone to contamination, why not collect samples such as oral or anal samples and expand the sampling area.

Blood is the preferred sample type for ASFV diagnosis (Beltran-Alcrudo et al., 2017; European Food Safety Authority, 2021). The field team was trained in proper sample collection, storage, and processing to minimize contamination risks. While oral and rectal swabs can be alternative sample types, they were not prioritized in this study due to ASFV’s higher detection reliability in blood.

References:

Beltran-Alcrudo, Daniel & Arias, Marisa & Gallardo, Carmina & Kramer, Scott & Penrith, Mary. (2017). African swine fever: detection and diagnosis – A manual for veterinarians.

EFSA (European Food Safety Authority), Gervelmeyer A, 2021. Public consultation on the draft data section on the ability of ASFV to survive and remain viable in different matrices of the Scientific opinion on Risk assessment of African swine fever and the ability of products or materials to present a risk to transmit ASF virus.

We hope these revisions address your concerns and enhance the overall quality of the manuscript; once again, thank you for your thoughtful review.

Kind regards,

Dr. Auer

Reviewer 2 Report

Comments and Suggestions for Authors

This study examines farming practices and the distribution of ASFV genotype II in Tanzania's Southern Highlands, with a particular emphasis on biosecurity gaps and molecular characterization. The molecular analysis demonstrated genetic uniformity among the isolates, all of which clustered within ASFV genotype II. This research highlights the urgent need for improved biosecurity measures and farmer education to mitigate ASFV outbreaks in endemic regions. Overall, this study presents clear logical structure and reliable data; however, several issues warrant further attention.

1. The Fig.1 is unclear; it is recommended to use a higher resolution image and provide labels.

2. Please present the detailed information of the samples in a table format.

3. Fig. 5 also needs to analyze whether the amino acid sequence has changed.

4. Whether the recombinant strains have been analyzed, it is recommended to analyze and discuss.

Author Response

Dear reviewer,

thank you for your insightful comments (1-4) and suggestions, which have helped us improve the clarity and quality of our study. We addressed your 

  1. The Fig.1 is unclear; it is recommended to use a higher resolution image and provide labels.

The Figure was changed accordingly.

  1. Please present the detailed information of the samples in a table format.

A table was included accordingly as Supplementary data 2.

  1. 5 also needs to analyze whether the amino acid sequence has changed.

The nucleotide substitution/deletion in ECOA sequences in the Tanzanian samples shown in Figure 5, are located in I73R-I329L intergenic spacer and not the coding regions, therefore there was no need to analysis the analyze the amino acid sequence. This was added in lines 217-219.

  1. Whether the recombinant strains have been analyzed, it is recommended to analyze and discuss.

The analysis of the Tanzanian ASFV 2024 isolates based on the 5 targeted genes clearly indicated that they are genotype II, and none of the genes classified the isolates in any other ASFV genotype which would imply recombination. This was added in lines 220-222.

We believe we have adequately addressed your comments (1-4), and we hope these revisions meet your expectations. Thank you again for your valuable review.

Dr. Auer

Reviewer 3 Report

Comments and Suggestions for Authors

The manuscript examines ASF in Tanzania, highlighting gaps in biosecurity and analyzing the phylogenetic relationships among Tanzanian strains. It provides a clear description of the epidemiological situation and presents the results effectively. However, a few minor points need to be addressed:

Line 111: Supplementary data could not be found.

Line 124: Auer et al. (2022) validated several kits. The authors should describe the methodology they used.

Line 136: The reference Chen et al. (2023) could not be found online. Additionally, the title appears to focus only on PCV2.

Line 137: The authors should consider that CSF is an RNA virus and ensure the methodology aligns accordingly.

Line 152: The reference is missing.

Line 161: Please specify which evolutionary models were used and the criteria for their selection.

Lines 288–291: The authors should reconsider the suitability of the spleen for testing these diseases. Specifically, the spleen is not an appropriate sample for detecting Salmonella, PPV, and PRV.

Author Response

Dear reviewer,

Thank you for your constructive comments and thoughtful suggestions, which have greatly contributed to enhancing the clarity and accuracy of our manuscript.

Line 111: Supplementary data could not be found.

S1 was added into the word document.

Line 124: Auer et al. (2022) validated several kits. The authors should describe the methodology they used.

The information was added accordingly as Auer et al (2022) analyses master mixes by assessing sensitivity, specificity, and reproducibility using ASFV-positive and ASFV-negative DNA samples, comparing performance based on Cq values, limit of detection, and concordance with reference methods. 

Line 136: The reference Chen et al. (2023) could not be found online. Additionally, the title appears to focus only on PCV2.

The reference was corrected.

Chen, Y., Luo, S., Tan, J., Zhang, L., Qiu, S., Hao, Z., Wang, N., Deng, Z., Wang, A., Yang, Q., Yang, Y., Wang, C., Zhan, Y., 2023. Establishment and application of multiplex real-time PCR for simultaneous detection of four viruses associated with porcine reproductive failure. Front. Microbiol. 14, 1092273 https://doi.org/ 10.3389/fmicb.2023.1092273

Line 137: The authors should consider that CSF is an RNA virus and ensure the methodology aligns accordingly.

Thank you for your helpful comment. We appreciate you pointing out the need to consider that CSF is an RNA virus. However, since only DNA was sent for analysis and not total nucleic acid to APHL, CSF was not detectable in this case. We have updated the methodology and results sections accordingly to reflect this clarification.

Line 152: The reference is missing.

Reference was added.

Line 161: Please specify which evolutionary models were used and the criteria for their selection.

The evolutionary models applied for each phylogenetic tree was included in the corresponding figures (Fig.3 and 4.) The p72 Neighbor-joining tree was inferred using with evolutionary distances computed using the Maximum Composite Likelihood method, the p54 tree was based on Minimum Evolution method and evolutionary distances computed using the Kimura 2‐parameter method, while the CD2v analysis was based on Maximum‐likelihood method computed using the General Reversible Chloroplast model with Gamma distribution.

The suitability of model for different datasets based on the average Jukes-Cantor (JC) distance which can be estimated in MEGA. Different phylogenetic approaches were applied for each dataset and compared with NJ tree. For instance, the average JC distance of the p72 dataset was 0.03, and the ML tree generated from this dataset collapsed some of the genotype branches, compared to NJ tree. Therefore, NJ was suitable for p72 sequence analysis.

Lines 288–291: The authors should reconsider the suitability of the spleen for testing these diseases. Specifically, the spleen is not an appropriate sample for detecting Salmonella, PPV, and PRV.

Thank you for your comment regarding the sample type used in the study. We would like to clarify that the samples analyzed were not spleen tissue, but whole blood, as described in the Methods section.

We hope that we have addressed your comments (1-7) comprehensively and that the revisions meet your expectations. Thank you once again for your valuable feedback.

Kind regards,

Dr. Auer

Reviewer 4 Report

Comments and Suggestions for Authors

I found Auer's manuscript very interesting and generally well-written.

I recommend its publication after addressing the only issue I identified.

In my opinion, a clock calibration (and accordingly a molecular dating) would be useful to accurately determine the temporal origin in the area of interest.

Author Response

Thank you for your suggestion regarding the time-scaled phylogeny. We appreciate your input, and we would like to clarify that the phylogenetic analysis in this study is based on the five targeted genes, not whole genome sequencing (WGS), as stated in the Methods section. Since WGS was not undertaken in this study, we did not perform clock calibration or molecular dating. The analysis of the Tanzanian ASFV 2024 isolates using these five targeted genes clearly indicated that they belong to genotype II. We hope this provides clarity on the methodology used in this study. 

Kind regards,

Dr. Auer